



# Implementing detailed nucleation predictions in the Earth system model EC-Earth3.3.4: sulfuric acid-ammonia nucleation

Carl Svenhag[1], Moa K. Sporre[1], Tinja Olenius[4], Daniel Yazgi[4], Sara M. Blichner[3], Lars .P Nieradzik[2], and Pontus Roldin[1]

[1]Department of Physics, Lund University, Lund, Sweden
[2]Department of Physical Geography, Lund University, Lund, Sweden
[3]Department of Environmental Science and Analytical Chemistry, Stockholm University, Stockholm, Sweden
[4]Swedish Meteorological and Hydrological Institute, Norrköping, Sweden

**Correspondence:** Carl Svenhag (carl.svenhag@forbrf.lth.se)

**Abstract.** Representing detailed atmospheric aerosol processes in global Earth system models (ESMs) has proven challenging both from a computational and a parameterization perspective. The representation of secondary organic aerosol (SOA) formation and new particle formation (NPF) in large ESMs are generally constructed with low detail to save computational costs. The simplification could result in losing the representation of some processes. In this study, we test and evaluate a new approach for improving the description of NPF processes in the ESM EC-Earth3 (ECE3) without loss of significant computational time. The current NPF scheme in EC-Earth3 is derived from the nucleation of low volatility organic vapors and sulfuric acid ($H_2SO_4$) together with a homogeneous $water-H_2SO_4$ nucleation scheme. We expand the existing schemes and introduce a new look-up table approach that incorporates detailed formation rate predictions by molecular modeling of sulfuric acid-ammonia nucleation ($H_2SO_2$-$NH_3$). We apply tables of particle formation rates for $H_2SO_2$-$NH_3$ nucleation, including dependence on temperature, atmospheric ion production rate, and molecular cluster scavenging sink. The resulting differences between using the $H_2SO_4-NH_3$ nucleation in ECE3 and the original default ECE3 scheme are evaluated and compared with a focus on changes in the aerosol composition, cloud properties, and radiation balance. From this new nucleation scheme, EC-Earth3's global average aerosol concentrations in the sub-100 nm sizes increased by 12 - 28 %. Aerosol concentrations above 100 nm and the direct radiative effect (in $Wm^{-2}$) from the changed nucleation only resulted in minor changes. However, the radiative effect from clouds affected by aerosols from the new nucleation scheme resulted in a global decrease (cooling effect) by 0.28 - 1 $Wm^{-2}$. Additionally, several stations with observed aerosol size number distribution measurements are compared with the model results to examine the performance of the NPF schemes.

## 1 Introduction

Understanding atmospheric particulate matter and its influence on the climate and air quality is a vital scientific question for the outcome of our future planet (Canadell et al., 2021). We generally categorize two types of aerosol particles in the atmosphere based on their emission pathway, either emitted directly as "primary aerosols" or formed indirectly from precursor gases in the air as "secondary aerosols". From a global climate perspective, the significance of new particle formation (NPF) leading



to secondary aerosols has been shown to be broader than previously believed (Merikanto et al., 2009; Dunne et al., 2016).
As aerosols in the atmosphere can scatter or absorb radiation in various wavelengths, additional secondary aerosols could
promote planetary cooling or warming through the direct aerosol radiative effect (DRE). Furthermore, the secondary aerosols
can influence the formation, properties, and lifetime of clouds, changing the reflective ability of clouds for incoming sunlight
radiation, potentially cooling the planet (Twomey, 1974; Albrecht, 1989).

Secondary aerosols can form by condensation of vapors on pre-existing particles or through new particle formation (NPF).
Studies on the radiative outcome and other climate effects caused secondary aerosols have developed in recent years but are still
highly uncertain (Shrivastava et al., 2017; Canadell et al., 2021). NPF occurs through gas-phase molecules forming molecular
clusters that grow further into larger particles by condensation of low-volatile vapors. While understanding of the chemical
species that drive the initial clustering processes has improved significantly during the last decade, the exact mechanisms and
their effects on a global scale continue to be highly uncertain. A challenge in understanding the future extent of secondary
aerosol climate effects is that the formation process in the atmosphere for aerosols is itself influenced by its ambient condi-
tions (e.g., temperature and humidity). Outlining the correct conditions for secondary particle formation globally is crucial to
quantify the various feedback mechanisms involved and the net future effects of climate change (CMIP6 2022).

Recent research indicates that low-volatility organic compounds (LVOCs) have an important role in the growth of aerosols
with sizes starting from 1 nm (Paasonen et al., 2010; Kirkby et al., 2011; Ehn et al., 2014; Riccobono et al., 2014; Tröstl et al.,
2016; Öström et al., 2017; Roldin et al., 2019). VOCs in the atmosphere can exist in many different molecular constructions,
and model estimates show that up to 85% of the VOCs originate from natural sources, labeled as (biogenic) BVOCs (Lamarque
et al., 2010; Guenther et al., 2012). In many Earth system models, VOCs are typically reduced to only two dominating species
categorized by their volatility: semi-volatile (SVOC) and extremely low-volatile (ELVOC) (Sporre et al., 2020). These BVOCs
are primarily formed by the oxidation of two naturally emitted precursors isoprene and monoterpene. Experimental studies
show that BVOCs can heavily influence the formation and growth of secondary organic aerosols (SOA) in the atmosphere
(Kulmala et al., 2004, 2013; Dunne et al., 2016). Most of the ambient BVOC gases will end up growing pre-existing particles
by condensation, but some may also contribute to NPF. However, the estimations of BVOC's net contribution to the global
SOA budget are not well understood (Tsigaridis et al., 2014; Shrivastava et al., 2017).

To derive global scale estimations for secondary particle formation and SOA budgets and their climate effects, we can
use the application of Earth-system models (ESMs). Many ESMs have parameterization for particle formation rates derived
exclusively from binary homogenous nucleation and condensation of atmospheric sulfuric acid ($H_2SO_4$) and water in the gas-
phase (Vehkamäki, 2002). This method has yielded general underestimations for modeled results in boundary layer aerosol
concentrations compared to observations (Mann et al., 2012). More recent model development has included the extremely
low-volatile organic compounds (ELVOCs) in the NPF schematics and chemistry with strong growth (survival) dependency
on the BVOCs (Kerminen and Kulmala, 2002; Bergman et al., 2021). However, experimental studies and detailed modeling
have shown that atmospheric gas-phase ammonia ($NH_3$) also plays an essential role in $H_2SO_4$-driven molecular clustering and
cluster growth (Dunne et al., 2016; Roldin et al., 2019). Ammonia is predominantly emitted from agricultural sources and is
not included in all ESM chemistry models, which obstructs its participation in the potential NPF schemes.



In this study, we use the ESM EC-Earth3 (ECE3) which includes atmospheric concentrations of ammonia. We implement a new scheme for ESM boundary layer NPF based on detailed modeling of molecular cluster formation kinetics with quantum-chemistry derived input data for cluster evaporation (Olenius et al., 2013). This high-level molecular modeling approach has become a standard tool in NPF studies and has been used for detailed representations of particle formation applied in previous box models and column model studies (Roldin et al., 2019; Wollesen de Jonge et al., 2021). In this study, we test and evaluate the global application of this approach by incorporating the detailed formation rate predictions through a lookup table interface (Yazgi and Olenius, 2023b). Due to the high computational load of running a molecular cluster simulation fully coupled with EC-Earth, we utilize this lookup table approach for optimal performance. The EC-Earth3 model version in this study is part of the Coupled Model Intercomparison Project (CMIP) and we wish to further evaluate and improve the EC-Earth3-AerChem configuration (van Noije et al., 2021). The previous nucleation rate scheme for NPF in EC-Earth3 is based on Riccobono et al. (2014) which approximates the rate as a function of gas-phase ELVOC and $H_2SO_4$ concentrations. In this study, we evaluate the previous scheme against the new lookup tables of $H_2SO_4-NH_3$ particle formation rates calculated using two different input quantum chemistry data sets for cluster evaporation. These two new table data sets are known to have tendencies towards under and over-predictions, and can thus be applied to assess the lower- and upper-limit effects of $H_2SO_4-NH_3$ nucleation. (Kürten et al., 2016; Besel et al., 2020). Since studies also support the mechanism of pure organic-$H_2SO_4$ (without $NH_3$) nucleation (Metzger et al., 2010; Riccobono et al., 2014), we include a fourth simulation with the lower-biased $H_2SO_4-NH_3$ nucleation scheme together with the default Riccobono ELVOC$-H_2SO_4$ nucleation. We will evaluate the resulting aerosol size number distributions from the four simulated EC-Earth3 schemes and compare them with observed measurements from multiple ground-based field stations. This study will also compare the resulting changes in the modeled cloud characteristics and radiative balance from using the new NPF scheme.

## 2 Model description

### 2.1 General

In this study, we use EC-Earth3.3.4 as the EC-Earth3-AerChem configuration, which includes the Global Circulation Model (GCM) Integrated Forecasting System (IFS) on cycle 36r4 coupled with chemistry from Tracer Model 5 Massively Parallel (TM5-MP) version 1.2 (Krol et al., 2005; van Noije et al., 2014; Williams et al., 2017). The IFS GCM includes the integrated land-surface model H-TESSEL (Balsamo et al., 2009). The models exchange information through the coupler OASIS3-MCT version 3.0 (Craig et al., 2017) with the coupling frequency between IFS and TM5 set to 6 hours. The IFS model time step was set to 45 minutes with instantaneous output averaged into 6-hourly values and TM5 had 1-hour time steps with monthly averaged output. The IFS meteorology was nudged against ERA-Interim divergence, vorticity (U and V winds), and surface pressure, with a relaxation time of 6 hours. The nudging of these parameters will force homogeneity in the general synoptic weather for all simulations. For the horizontal resolutions, IFS operated on a T255 (0.7°) spectral truncation with an N128 reduced Gaussian grid and TM5 on a 3°x 2° (longitude x latitude) grid. Vertically, IFS and TM5 utilize the same represented hybrid sigma pressure levels, where IFS operates on 91 layers, while TM5 uses a lower resolution of 34 layers (excluding



the top IFS layer). A more detailed coupling description is given in van Noije et al. (2014, 2021) including information on the AMIP reader for the ocean interface. A known issue concerning the exclusion of atmospheric MSA for the EC-Earth3.3.4 version was corrected in this study.

## 2.2 Aerosol module M7 in TM5

The TM5-MP model represents the aerosol mass and number concentrations in the M7 module as seven log-normal modes (Vignati et al., 2004), with four "mixed" water-soluble nucleation (NUS), Aitken (AIS), accumulation (ASC), and coarse modes (COS), and three insoluble modes of Aitken (AII), accumulation (ACI), and coarse (COI) sizes. The aerosol log-normal distribution has fixed standard deviations and dry-radius size ranges given as: $r_{nucl} < 5$ nm, $5 < r_{Aitken} < 50$ nm, $50 < r_{accu} < 500$ nm, $r_{coarse} > 500$ nm. The six categorized species distributed (variously) over the seven modes are sea salt (SS), dust (DU),
black carbon (BC), sulfate (SO4), primary organic aerosol (POA), and secondary organic aerosols (SOA). For the water-soluble accumulation mode, there is additional condensation of methane sulfonic acid (MSA) and ammonium nitrate (AN), which can alter the optical properties and mass of the soluble accumulation mode. The optical characteristics of each species in the model are described by Mie theory look-up tables (Aan de Brugh, 2013). For more details on M7 aerosol modal dynamics and species, see Vignati et al. (2004) and van Noije et al. (2014).

## 2.3 Secondary Aerosol formation

The TM5-MP chemistry in EC-Earth3 uses two BVOC emission species for non-methane VOCs that oxidize in the chemistry scheme: monoterpene ($C_{10}H_{16}$) and isoprene ($C_5H_8$). The two BVOCs have prescribed model inputs from monthly 0.5° x 0.5° emissions based on the MEGAN-MACC inventory (Sindelarova et al., 2014). The monthly emissions are then balanced in TM5 to a diurnal distribution formula for the 1-hour time step. Subsequently, the BVOCs are oxidized from specified reaction
yields with ozone ($O_3$) or hydroxyl radicals (OH) into SVOC ($C_{10}H_{16}O_6$) or ELVOC ($C_{10}H_{16}O_7$). Rate coefficients are based on (Atkinson et al., 2006) and the molar yields for producing ELVOCs and SVOCs are tabulated in Bergman et al. (2021). Both VOC groups can condense to the three larger soluble modes and the insoluble Aitken mode. Furthermore, the ELVOCs are included in the default NPF scheme and for the growth of nucleated particles to 5 nm in diameter through condensation in all schemes (including the CLUST cases) through the Kerminen and Kulmala (KK) factor of survival (Kerminen and Kulmala,
2002). There are two nucleation rates in the default NPF scheme in TM5. The first $J_{Riccobono}$ is the nucleation based on Eq. 1 from semi-empirical Riccobono et al. (2014) parameterization for particle formation rate at 1.7 nm diameter:

$$J_{Riccobono} = K_m \left[H_2SO_4\right]^2 \left[ELVOC\right] \tag{1}$$

$K_m = 3.27 \times 10^{-21}$ cm$^6$ s$^{-1}$ is a constant empirical factor and the two species [$H_2SO_4$] and [ELVOC] represents the gas-phase concentrations. The second nucleation rate in TM5 is the binary homogeneous nucleation (BHN) of water and $H_2SO_4$
following Vehkamäki (2002). The BHN pathway is included in the configurations with the new $H_2SO_4-NH_3$ scheme. The KK factor is used to obtain the fraction of particles surviving growth to 5 nm for all the schemes ($H_2SO_4-H_2O$, $H_2SO_4-NH_3$, and $H_2SO_4-ELVOC$, for which the initial nucleation rates are given at sizes of ca. 1.0, 1.1, and 1.7 nm in diameter, respec-



tively) by condensational growth from available $H_2SO_4$ and ELVOC. The 5 nm particles are then partitioned into the modal system of M7. See the schematic figure for the representation of the initial growth in Fig. A1. The growth to 5 nm and the
formation of any new particles in TM5 is thereby limited by the available gas phase ELVOC and $H_2SO_4$, if the concentration of one compound is insufficient the other compound (if available) will account for the remaining growth to 5 nm diameter. The full growth parameterisation is given in Bergman et al. (2021).

## 2.4 Radiation and cloud interactions for aerosols

The activation of cloud droplets from aerosols is described by the activation scheme in Abdul-Razzak and Ghan (2000) which is a specific parameterization for modal aerosol models such as TM5-MP (M7). The soluble aerosol mode properties and supersaturation (derived from updraft velocity), determine the cloud droplet number concentrations (CDNC) of stratiform clouds in IFS, where the CDNC has a minimum of $30 \, \mathrm{cm^{-3}}$ (van Noije et al., 2021). The effective liquid droplet radii are subsequently determined by the CDNC and the liquid water content in-cloud from the activation scheme in Martin et al. (1994) and have a
radius set between 4 - 30 μm. The cloud-lifetime effect is then determined by these parameters following the autoconversion of liquid cloud droplets to rain. The effective cloud radius is used for the calculation of the cloud radiative scattering in each IFS model grid, see further description in van Noije et al. (2021) and Wyser et al. (2020).

### 2.4.1 Lookup tables of $H_2SO_4-NH_3$ nucleation rates

We implement a new look-up table approach to incorporate particle formation rates from molecular modeling by applying the J-GAIN tool (Formation rate look-up table Generator And Interpolator; Yazgi and Olenius 2023b, 2021), which includes automatic routines for table generation and interpolation. The table generator calculates formation rates by molecular cluster dynamics modeling through an embedded application of the ACDC (Atmospheric Cluster Dynamics Code) cluster kinetics solver (Olenius, 2021). Tables are generated for user-defined input for the chemical species and the ambient conditions that
determine the rate, including e.g. the concentrations of the precursor vapors and the temperature (for more details, see Yazgi and Olenius 2023b). This yields high-resolution formation rate data over a wide spectrum of atmospheric conditions. The chemistry input includes cluster compositions and quantum chemical thermodynamics data for calculating cluster evaporation (Elm et al., 2020; Olenius et al., 2013). The table interpolator applies multivariate interpolation to determine formation rates for given ambient conditions from user-defined tables. In this work, we generate formation rate tables for sulfuric acid and
ammonia ($H_2SO_4-NH_3$), which is a globally significant particle formation mechanism according to current understanding (e.g. Gordon et al., 2017). The rates are calculated as a function of [$H_2SO_4$], [$NH_3$], temperature, cluster scavenging sink (CS), and atmospheric ion production rate (IPR), considering both electrically neutral and ion-mediated pathways (as detailed in e.g. Olenius et al. 2013). In order to assess uncertainties related to the quantitative formation rate predictions, we use two alternative data sets computed with different quantum chemistry methods: a recent data set by the state-of-the-art method
DLPNO-CCSD(T)/aug-cc-pVTZ//$\omega$B97X-D/6-31++G(d,p), here referred to as CLUST-Low (Besel et al., 2020)), and a previ-





**Table 1.** The $H_2SO_4$ and $NH_3$ nucleation rate lookup-table ranges for variables used in this study. Values outside the $H_2SO_4$ and $NH_3$ lower limits return a zero nucleation rate, and the other three variables return the max or min value given here if limits are exceeded.

|  | $H_2SO_4$ [cm$^{-3}$] | $NH_3$ [cm$^{-3}$] | Temperature [K] | CS [s$^{-1}$] | IPR [cm$^{-3}$s$^{-1}$] |
|---|---|---|---|---|---|
| Lower limit: | $1 \times 10^5$ | $1 \times 10^6$ | 180 | $1 \times 10^{-5}$ | 0.1 |
| Upper limit: | $1 \times 10^8$ | $3 \times 10^{11}$ | 320 | $1 \times 10^{-1}$ | 60 |

ous data set by the RICC2/aug-cc-pV(T+d)Z//B3LYP/CBSB7 method, here referred to as CLUST-High (Olenius et al., 2013). These data sets can be expected to provide a realistic range for the predictions, as the CLUST-Low (DLPNO) method may under-predict the quantitative rate values, while CLUST-High (RICC2) has a tendency towards over-prediction (Besel et al., 2020; Kürten et al., 2016; Carlsson et al., 2020). We couple the table interpolator to the TM5 component and conduct simulations with either CLUST-High or CLUST-Low-based formation rates for ECE3. The $H_2SO_4$ and $NH_3$ concentrations covered by the lookup table have a restricted range (Table 1) where the routine returns zero nucleation rate if one or both of the concentrations are below the limits. For the input IPR, CS, and temperature, values are set to the maximum or minimum table value if the limits are exceeded. The wide value ranges are set to cover the variety of global conditions in the simulations.

The ion-pair production (IPR) from galactic cosmic rays (GCR) is determined from an additional two-parameter lookup table based on **?** added to the TM5 module for this study. This table reads the model pressure (203 layers) and magnetic latitude (91 latitudes) and calculates GCR based on calculations from Usokin and Kovaltsov (2006). The IPR resulting from soil radon in this function is calculated from the model land fraction and altitude, adapted from **?**.

## 2.5 Simulations

We include four separate simulations for EC-Earth3.3.4 over a five-year period from 2014 - 2018 with a one-year spin-up period. The four simulations in this study are referred to as (1) control, (2) CLUST-High, (3) CLUST-Low, and (4) CLUST-Low+Riccobono. The control case is run with the default setup for EC-Earth3-AERCHEM with the nucleation rate based on Riccobono et al. (2014). For the two CLUST (High and Low) cases, we have replaced the Riccobono et al. (2014) based nucleation scheme with the CLUST lookup table function. The CLUST-High represents the use of the RICC2 version of the lookup table and CLUST-Low is the DLPNO version described in Section 2.4.1. The fourth simulation is set up with the CLUST-Low ($H_2SO_4-NH_3$) table nucleation rate coupled with the default Riccobono et al. (2014) nucleation rate from ELVOC$-H_2SO_4$ nucleation.

## 2.6 Ground station observations

For observation data in this study, we used the EBAS online data service for retrieving data sets of particle concentrations at measurement stations (Tørseth et al., 2012; Franco et al., 2022). The majority of these station datasets are situated in Europe and coverage outside this region is scarce. The particle number size distribution is measured using SMPS and DMPS instrumen-





tation and they are averaged to monthly mean values for uniformity with the model output. The measured minimum diameter sizes are limited to around 10 nm for the particle samplers, with an exception for SMEAR II and Hyltemossa station (3 nm). The amount of EBAS data within the 2014 - 2018 period is moderate and a tabulated description of the station measurements is given in Table A1. The selected stations in this study were chosen in order to obtain aerosol concentrations at different marine, urban, and rural environments at various altitudes.

## 2.7  Model post-process methods

As mentioned above, the IFS model output is a monthly average. For the IFS cloud characteristics: CDNC and effective liquid radius ($r_{eff}$) we apply a monthly weighted average for the 6-hourly output with respect to "cloud time" (IFS output variable) in each individual grid cell. This weighted average accounts for the actual lifetime of clouds in IFS spatially and temporally. For the IFS simulations, we use the function: "double call to radiation" diagnostics (Hogan and Bozzo, 2018), which gives two separate radiative fluxes with (and without) an "aerosol-free" atmosphere for calculating radiative differences following Ghan (2013). We can then represent the top-of-atmosphere (TOA) net (short plus long wave) radiative flux only influenced by aerosols in a "clear-sky" model environment, here referred to as the direct aerosol radiative effect (DRE). Similarly, the cloud radiative effect (CRE) is calculated from the "clear-sky" (aerosol-free) condition subtracted from the "all-sky" (aerosol-free) condition. See van Noije et al. (2021) for tabulated optical properties of all aerosol species in EC-Earth3 used for the radiative fluxes. In this study, for results classified as "near-surface", we use a weighted pressure level average to represent the bottom three model layers from TM5 output (chemistry and aerosol output). For "near-surface" values from the IFS output of CDNC and $r_{eff}$ we use an average (weighted) of data points below an 850 hPa cutoff layer.

## 3  Results and discussion

The resulting near-surface mean particle formation rate (number of 5 nm diameter particles formed per unit volume and time) for the four EC-Earth simulations is shown in Fig. 1. The highest formation rate for all cases occurs in the regions with anthropogenic influence, with the greatest values in south and eastern Asia. The CLUST-Low+Riccobono case in Fig. 1d shows the resulting particle formation rate using both $H_2SO_4-NH_3$ nucleation and $ELVOC-H_2SO_4$ nucleation. The CLUST-High nucleation scheme has the highest mean particle formation rate, and compared with the default control case it gives increased rates at higher latitudes, but lower rates in the tropical regions. Some tropical regions have higher BVOC concentrations with lower $NH_3$ and $H_2SO_4$ concentrations, so here $ELVOC-H_2SO_4$ near-surface nucleation is dominating in the model. The CLUST scheme cases introduce near-surface particle formation over the ocean from $H_2SO_4-NH_3$ nucleation seen in Fig. 1). This was negligible ($< 10^{-4}$) in the previous default model scheme due to the absence of marine ELVOCs and low BHN. However, ECE3 has gas-phase ammonia and sulphuric acid present in these marine regions as ammonia can be transported from land airmasses and primary marine emissions sources, which gives boundary layer $H_2SO_4-NH_3$ nucleation from the CLUST scheme.



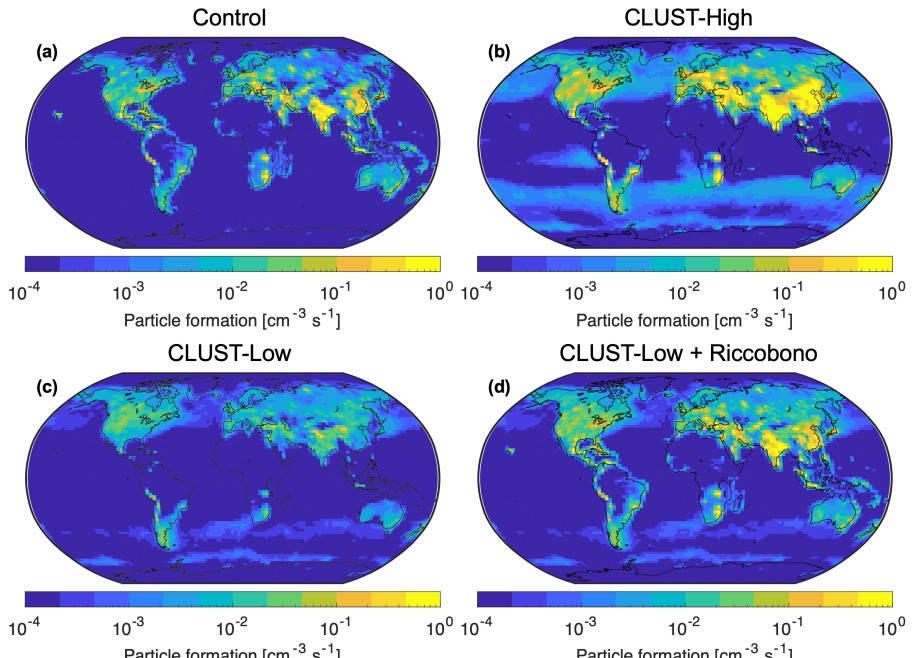

**Figure 1.** The 5-year mean of 5 nm diameter aerosol particle formation rate (post KK survivability) for the control run (**a**), the CLUST-High case (**b**), the CLUST-Low (**c**), and CLUST-Low+Riccobono (**d**) at near-surface level.

## 3.1 Global aerosol concentrations

The global mean vertical profiles for aerosol number concentrations for the four simulations can be seen in Fig. 2, where CLUST-High produces the highest particle formation rate profile and subsequently results in the highest aerosol number con-
centrations in the nucleation and Aitken modes. The mean profiles of CLUST-Low and CLUST-Low+Riccobono's soluble nucleation (NUS) and Aitken (AIS) mode aerosol number concentrations are substantially lower than CLUST-High but still higher than the control run. Fig. 2c shows that the modeled global mean soluble accumulation mode gives similar mean profile values for all four simulations and has a minor response to the altered nucleation scheme. Furthermore, in the upper troposphere and lower stratosphere, the BHN from water and $H_2SO_4$ is dominating, which results in less modeled differences for the par-
ticle formation rates and the aerosol concentrations at these altitudes. The resulting change in the global average concentration (total atmosphere) of sub-100 nm aerosols was an increase of 27.8 % for CLUST-High, 11.7 % for CLUST-Low, and 12.6 % for CLUST-Low+Riccobono. In the surface layer, the mean particle formation rate in CLUST-Low is lower compared to the control run (by a factor of 10) but has higher nucleation mode concentration. This is likely due to more aerosols being transported down from the overlying model layers where CLUST-Low has greater particle formation. The nucleation mode
could also be reduced if the control case experiences regional removal effects (e.g. high heterogeneous coagulation) that differ from the CLUST-Low. Fig. A4 and Fig. A5 show the different global NUS and AIS concentrations at near-surface for the four simulations.



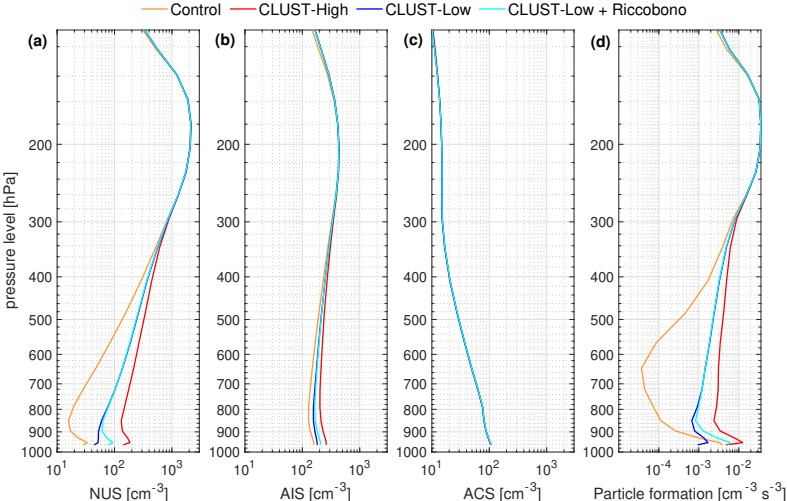

**Figure 2.** Modeled global mean profile for aerosol number concentrations for the soluble nucleation (NUS), Aitken (AIS), accumulation (ACS) mode, and particle formation rate of 5 nm aerosols.

The vertical profile in Fig. 2d shows the significant increase in the global mean particle formation rate in the free troposphere for the three new CLUST-scheme simulations. This could be explained by three potential changes made in our new

parameterization. Firstly, the introduced dependency for the nucleation rate in CLUST to ion-pair production which increases with altitude. Secondly, the effect of decreasing temperature with altitude leads to an increasing nucleation rate from the CLUST lookup table as the default scheme is not temperature dependent. Lastly, the rate of decreasing atmospheric ammonia concentrations with altitude is lower than the decreasing rate of ELVOC concentrations in the model shown in Fig. A2.

Fig. 3 shows the relative and absolute differences in the sub 100 nm aerosol concentrations between the CLUST schemes

and the default control. The zonal mean shows a global aerosol increase in all cases with the exception of a tropical decrease in the lower troposphere for the CLUST-Low case. The most dominant zonal increases in the Northern hemisphere and mid-latitudes were expected for the absolute difference in Fig. 3a,b,c as they are the dominant regions of anthropogenic emissions of $NH_3$ and $H_2SO_4$. The vertical distribution of sub 100 nm aerosols shows the difference follows a similar zonal pattern, with the exception of CLUST-High differences (Fig. 3d) where a spike difference is occurring in the 800 - 500 hPa layer.



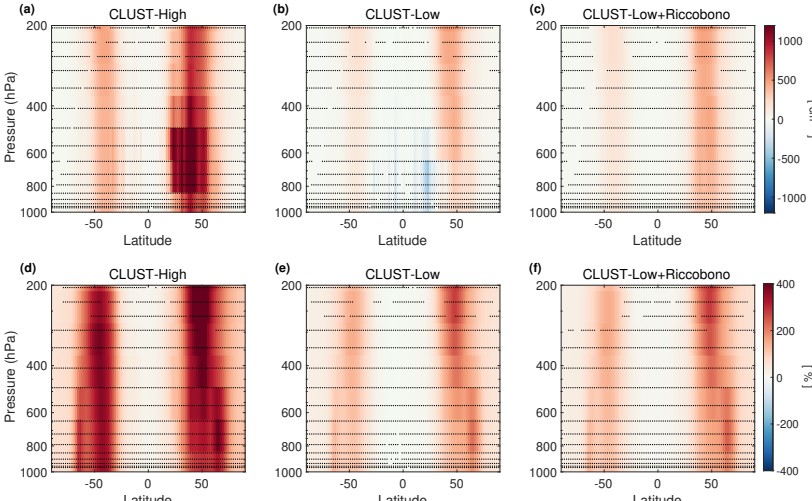

**Figure 3.** The CLUST cases - control run difference of the zonal mean profile for sub 100 nm aerosol number concentrations (total sum of NUS, AIS, and AII modes), with the absolute difference **(a)**, **(b)**, and **(c)**, and the relative difference **(d)**, **(e)**, and **(f)**, showing Student $t$ test significance as dots.

## 3.2 Modeled station observations

Fig. 4 shows the four model setup outcomes at twelve different station measurement locations. Similar to the global mean, the CLUST-High case consistently produces the highest sub-100 nm particles at all simulated stations. For all four model setups, the accumulation mode (> 100 nm) concentrations at the stations remain similar (matching Fig. 2). For nine out of twelve stations in Fig. 4 the model underestimates the Aitken mode concentrations with exceptions for the CLUST-High case at the SMEAR II, Aspvreten, Izaña, and Storm Peak. At these four stations, the CLUST-High case has good agreement with the measured concentrations in the Aitken mode. All CLUST cases have better agreement with the measurements at these four stations. The station settings at SMEAR II and Aspvreten are rural forests, while Izana and Storm Peak are high-altitude mountain settings. Two exceptions where the default schemes are close to the measured observations are found in the Arctic station on Svalbard (Fig. 4f) and the mountain station in the Swizz Alps at a 3454 m altitude (Fig. 4l). At the Amazonian ATTO station, the difference between CLUST cases and the control for the model mean aerosol concentrations is very small. Both model schemes likely produce very little NPF in this tropical region due to the absence of $H_2SO_4$.

The underestimated Aitken and accumulation mode we see across the urban stations have three potential causes: (1) An underestimation in the modeled primary emissions of particles. (2), An underestimation of available condenseable vapors or model restrictions in aerosol growth through condensation, suggested by (Bergman et al., 2021). (3) The low resolution (3°x 2° ) grids in TM5 we use for the station interpolation may differ significantly from the local station conditions, especially at urban stations. In the same way, an overestimation from extrapolating local conditions can be true for the modeled nucleation and Aitken mode aerosols at Jungfraujoch in Fig. 4l, as this grid-box covers a large central European region with high $H_2SO_4$ and



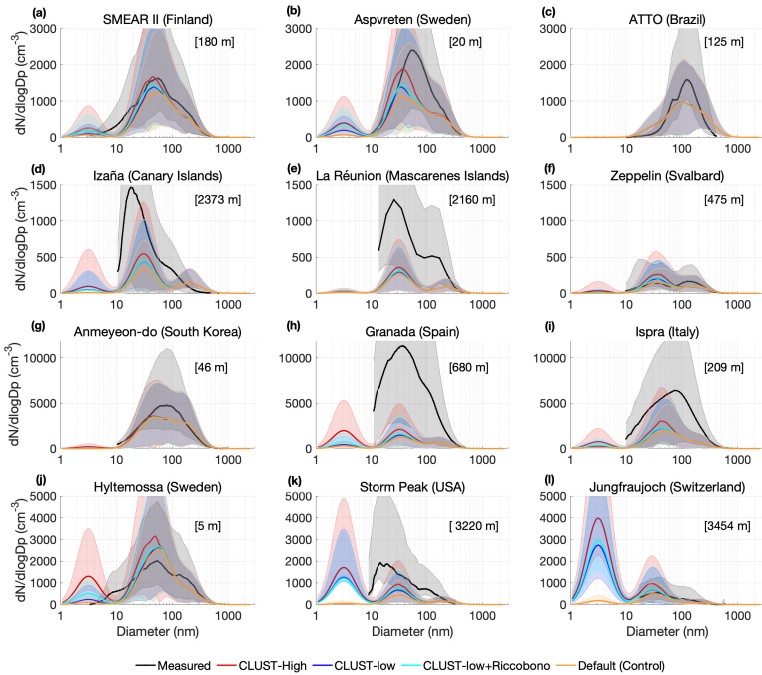

**Figure 4.** Stations of DMPS/SMPS comparison for median aerosol size number distributions at different locations. The shaded area shows the 25 percentile and the measurement altitude is given in each graph. For a full station description see Table A1.

$NH_3$ emissions that enable high particle formation rates in the CLUST scheme. The extreme high-altitude difference between model results for the median nucleation mode concentrations seen at Jungfraujoch and Storm Peak (Fig. 4k-l) is discussed

further in Section 3.5.

Another limitation in this model-observation comparison is the SMPS's and DMPS's cutoff diameter at ∼10 nm where the modeled M7 nucleation mode begins, preventing us from evaluating the model performance for the smaller mode concentrations. Additionally, measurement uncertainty may be higher close to this cutoff diameter. This could explain the dramatic decrease in the observed concentrations of just above 10 nm diameter concentrations at the Izaña station (Fig. 4d).

**3.3 Cloud properties' changes**

Figure 5 shows the difference for mean CDNC and liquid cloud droplet effective radius ($r_{eff}$) between the ECE3-CLUST cases and the control run. The mean CDNC increases significantly (t-test) for all three cases and extremes are found above the mid-latitudes in the North American continent and the Atlantic Ocean. Correlating to the most extreme regions for the particle formation in Fig. 1 and NUS (AIS) concentrations in Fig. A4 (AIS-figure), the resulting increase of particle formation

and subsequent AIS concentrations over the North American region could relate to a strong sensitivity for the changing cloud properties locally and downwind seen in Fig. 5. The highest extreme regions for particle formation and NUS concentrations





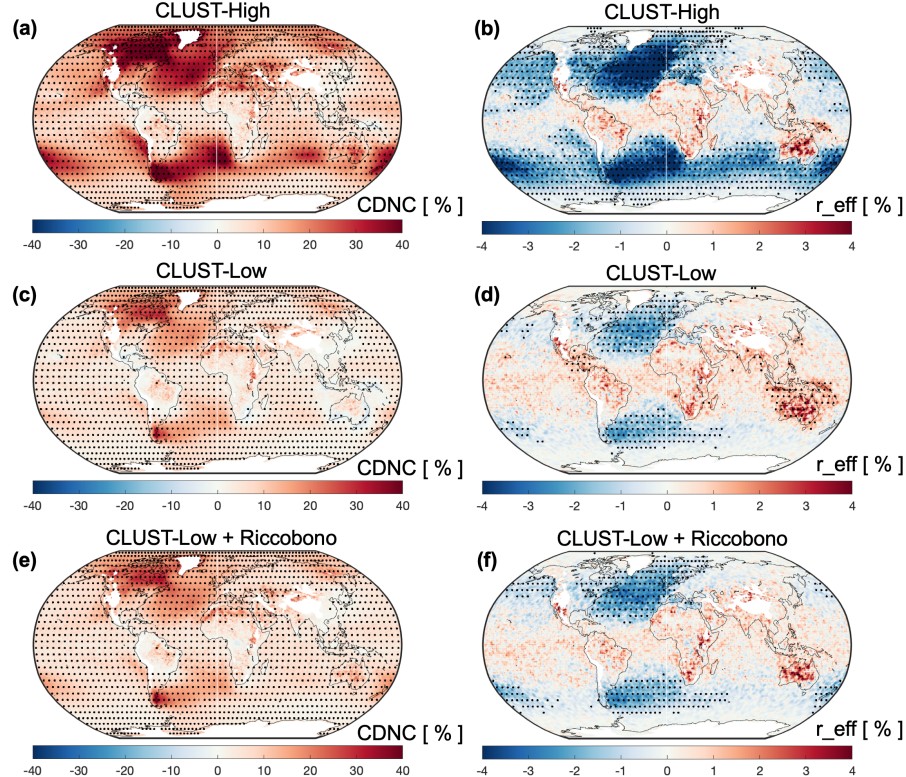

**Figure 5.** The global near-surface (> 850 hPa) mean liquid CDNC and cloud effective radius ($r_{eff}$) as a difference between the default control case and the CLUST-High (a, b), CLUST-Low (c, d), and CLUST-Low+Riccobono (e, f). Resulting differences computed with Student *t* test are shown as dotted regions with a 95 % significance.

over India and China show minor significant relative change for CDNC and cloud effective radius, and clouds show less sensitivity to sub-100 nm aerosol changes in these regions. The global CDNC concentrations increased by 12,1 % for CLUST-High, 5.9 % for CLUST-Low, and 6.7 % for CLUST-Low+Riccobono. Furthermore, the global effective liquid cloud radius decreased by -0.41 % for CLUST-High, -0,04 % for CLUST-Low, and -0.13 % for CLUST-Low+Riccobono.

### 3.4 Radiative responses

The net direct radiative effect (DRE) resulted in a small negative forcing from the elevated global particle formation with most net negative RF in CLUST-High compared to CLUST-Low and CLUST-Low+Riccobono (Fig. 6 left column). The global DRE changed by -0.010 W m$^{-2}$ for CLUST-High, 0.002 W m$^{-2}$ for CLUST-Low, and 0.008 W m$^{-2}$ for CLUST-Low+Riccobono. The EC-Earth3 cloud radiative effects (CRE) shown in Fig. 6 (right column) is highly sensitive to changes in the sub-100 nm aerosol number concentrations (Sporre et al., 2020). All CLUST cases with the new method of modeled nucleation rates



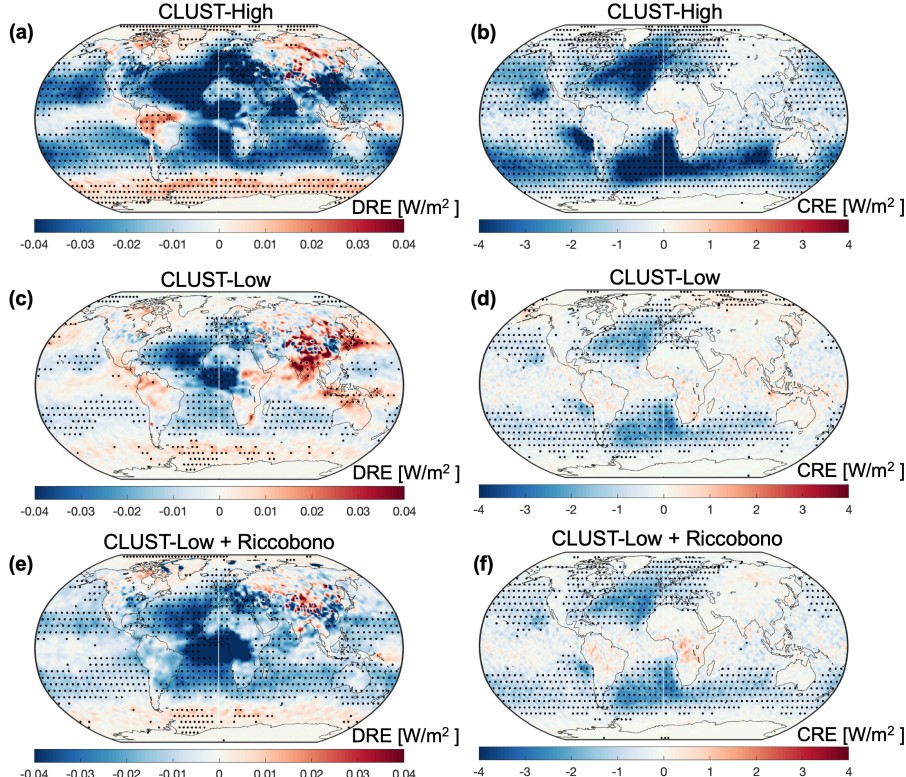

**Figure 6.** The global mean net TOA downward radiation modeled difference for the direct aerosol effect (DRE; a, c, e), and cloud radiative effect (CRE; b, d, f).

resulted in strong global negative CRE changes from the $H_2SO_4-NH_3$ scheme. The global mean CRE changed by -1.03 W m$^{-2}$ for CLUST-High, -0.28 W m$^{-2}$ for CLUST-Low, and -0.42 W m$^{-2}$ for CLUST-Low+Riccobono. The strongest

common negative RF occurrences occur over the oceans, and the negative RF over the North and South Atlantic are most prominent for both DRE and the CRE for all three CLUST simulations. This coincides with the results in Fig. 5 which shows a raised concentration of CDNC and decreased effective liquid radius over these regions. For the CLUST-High CRE case in Fig. 6b the marine stratiform cloud region exceeds negative 4 W m$^{-2}$ with the highest sensitivity to CDNC changes. As expected, the resulting scale between the direct aerosol and cloud radiative effects in Fig. 6 (left and right column) differ in magnitude

($\sim 10^2$) as DRE is governed mainly by aerosol scattering and absorption from the accumulation and coarse mode particles. These larger > 100 nm aerosols have less variation throughout the whole atmosphere with the new CLUST schemes (Fig. A6) in contrast to the sub-100 nm particle number concentrations (Fig. 3), which consequently will impact the CRE more than the DRE.



### 3.5 Further discussion

The difference between the control case and the CLUST schemes model outcomes for CDNC, effective cloud radius (Fig. 5), and cloud radiative effects (Fig. 6) demonstrate the high cloud and climate sensitivity to the M7 particle formation rate within EC-Earth3. This outcome is similar to the findings in Sporre et al. (2020), which state that EC-Earth3's "cleaner" atmosphere (compared to other ESMs), with fewer large particle concentrations (accumulation and coarse mode), gives greater CCN concentrations at higher sub-100 nm aerosol concentrations. The sub-100 nm aerosols in a "cleaner" atmosphere will

not be lost by coagulation by present larger particles. This can consequently increase the total number of aerosols that can act as CCN in the EC-Earth3 model compared to other ESMs (Sporre et al., 2020). Interestingly, the CRE outcome from one case in the Sporre et al. (2020) study is similar to our results using the CLUST scheme. Their model results "No isoprene" yielded a global mean CRE increase of -0.82 W m$^{-2}$ as a result of the increasing mean near-surface sub-100 nm aerosol concentrations by $\sim$ 15 %, and our results for CLUST-High showed a -1.03 W m$^{-2}$ CRE difference from the control case after the sub-100

nm aerosol concentration increased by 27.8 %.

The combination of both modeled $H_2SO_4-NH_3$ and $ELVOC-H_2SO_4$ nucleation approaches (CLUST-Low+Riccobono) is considered the most theoretically accurate NPF description as all of these species have been shown to contribute to NPF processes by mentioned chamber measurements and modeling studies (Dunne et al., 2016; Roldin et al., 2019). The condensation of ELVOCs is included in the particle growth from 1.07 nm to 5 nm in the KK formula for our CLUST lookup table

simulations. However, the $H_2SO_4-NH_3$ pathway produces negligible or zero formation rates at conditions where $NH_3$ concentrations are very low. Therefore, the model runs without the $ELVOC-H_2SO_4$ pathway may give unrealistically low or even erroneously zero formation rates at low $NH_3$ and high ELVOC concentrations. Including both organic and $NH_3$ pathways for nucleation is more realistic considering the current understanding. Additionally, the CLUST lookup table limits we set for $NH_3$, $H_2SO_4$, and the other input variables for the lookup table can be modified if needed, and further diagnostics on this can

be made for future studies.

Our results show that the current default nucleation in EC-Earth3 has a tendency to underestimate the modeled aerosol concentrations compared with measured stations (Fig. 4). The CLUST cases show closer agreement with the measurements at stations where the model previously underestimated the aerosol concentrations. The high-altitude median nucleation mode concentrations modeled in EC-Earth3 for Jungfraujoch station (Fig. 4l) are predominantly higher for the $H_2SO_4-NH_3$ CLUST

scheme. This is a profoundly anthropogenic-influenced grid with high concentrations of $H_2SO_4$ and $NH_3$ rising from the surface grid beneath, and with lower temperatures aloft, this gives more particles from the CLUST lookup table.

A potential underestimation of modeled primary emissions, in EC-Earth3 could contribute to the low concentrations in the model compared to observations, but evaluating primary emission inventories is outside the scope of this study. Further evaluating the conditions set in the M7 model module regarding aerosol growth and the available condensable vapors is a

point of interest for our future ECE model development. Introducing ammonium nitrate as an available condensable vapor to the Aitken and nucleation mode (which now only exists for accumulation mode in EC-Earth3) could increase the growth and survivability of smaller particles.



## 4 Conclusions

A new approach for new-particle formation rates has been implemented in the chemistry module TM5-MP of EC-Earth3
using a lookup-table approach and molecular cluster formation modeling (CLUST). This introduces a detailed $H_2SO_4-NH_3$
nucleation which can be added to the existing $ELVOC-H_2SO_4$ scheme based on Riccobono et al. (2014). The $H_2SO_4-NH_3$
nucleation is a unique implementation for large ESMs but it is supported by theory, chamber experiments, and regional model
studies. Three five-year simulations using the CLUST lookup table were compared towards a control case with relative and
absolute differences for radiative forcing, cloud properties, and aerosol concentrations.

This study showed that the updates in the nucleation rate scheme in the M7 aerosol module (TM5-MP) in EC-Earth3 gave
significant differences in the results for the sub-100 nm aerosol concentrations and the model radiative effects. The introduction
of $NH_3-H_2SO_4$ nucleation in EC-Earth3 had the highest net impact on the free troposphere particle formation rates and
the sub-100 nm aerosol concentrations. The global average (total atmosphere) sub-100 nm aerosol increased by 27.8 % for
CLUST-High, 11.7 % for CLUST-Low, and 12.6 % for CLUST-Low+Riccobono. Consequently, the resulting CRE for all
CLUST cases gave an increased negative net TOA downward radiation with -1.03 W m$^{-2}$ for CLUST-High, -0.28 W m$^{-2}$ for
CLUST-Low, and -0.42 W m$^{-2}$ for CLUST-Low+Riccobono). Comparatively, the modeled >100 nm aerosol concentrations
and the resulting DRE had minor changes from the implemented nucleation scheme. Annual medians of measured station
DMPS/SMPS observations at nine measurement sites were compared against the four model results at various locations.
The model performed well in reproducing the Aitken mode number concentration at most of the station locations. At three
locations the model over-predicts the Aitken mode, here the default control case gives a closer representation of the observed
concentration. At the other nine (under-predicted) locations, CLUST-High has the best Aitken mode representation, and the
other two CLUST cases have a better representation compared to the control case. The CLUST-High and CLUST-Low schemes
were qualitatively consistent at all locations and for the global mean, which gives confidence that the modeled upper and lower
limits of $H_2SO_4-NH_3$ nucleation follow the general trend.



**Appendix A**

**Table A1.** The station descriptions for all the observed measurements used in this study (Tørseth et al., 2012; Franco et al., 2022).

| Station Name | Location | Instrument | Data Time Period | Lat °N | Lon °E | Altitude | Setting |
|---|---|---|---|---|---|---|---|
| SMEAR II | Finland | DMPS | 14/01/01-18/12/31 | 61.84 | 24.29 | 180 m | Forest/Rural |
| Aspvreten | Sweden | DMPS | 14/01/01-14/12/31 | 58.81 | 17.38 | 20 m | Forest/Coastal |
| Hyltemossa | Sweden | DMPS | 18/01/01-18/12/31 | 56.10 | 13.42 | 5 m | Forest/Rural |
| La Réunion | Mascarenes | SMPS | 17/01/01-18/12/31 | -21.08 | 55.38 | 2160 m | Mountain/Island |
| Izaña | Tenerife | SMPS | 14/01/01-14/12/31 | 28.31 | -16.50 | 2373 m | Mountain/Island |
| Zeppelin | Svalbard | DMPS | 16/01/01-17/12/31 | 78.91 | 11.88 | 475 m | Polar/Island |
| Anmyeon-do | South Korea | SMPS | 17/07/01-18/06/31 | 36.54 | 126.3 | 46 m | Agricultural |
| Granada | Spain | SMPS | 17/01/01-17/12/31 | 37.16 | -3.61 | 680 m | Urban |
| ATTO | Brazil | SMPS | 14/01/01-18/12/31 | 45.80 | 8.63 | 209 m | Rural/Forest |
| Storm Peak | USA | SMPS | 14/01/01-18/12/31 | 40.45 | -106.74 | 3220 m | Mountain |
| Jungfraujoch | Switzerland | SMPS | 17/01/01-18/12/31 | 46.55 | 7.99 | 3578 m | Mountain |





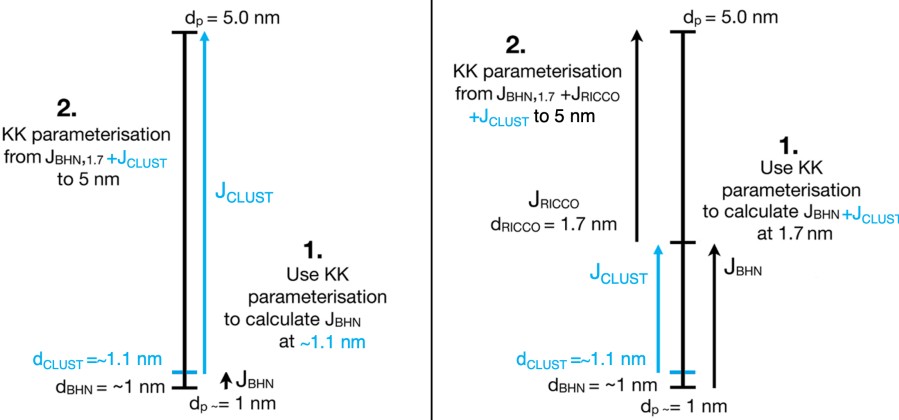

**Figure A1.** Schematic presentation of the parameterized growth of nucleated particles to the size of 5 nm. The left illustration shows the approach for the Clust-Low and Clust-High simulations. and the right illustration depicts the scaling within Clust-Low+Riccobono simulations. For the default control set-up; subtract the cyan colors from the right illustration.



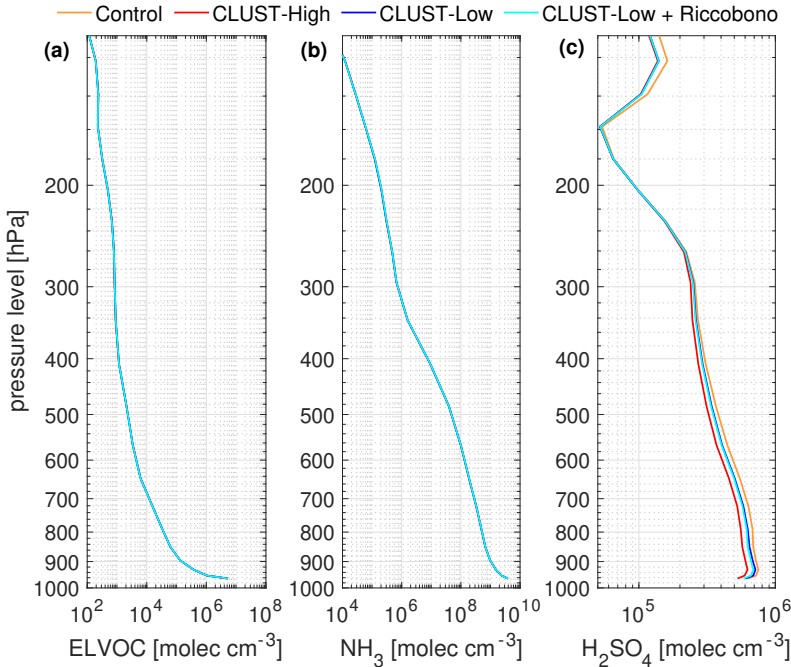

**Figure A2.** The global mean vertical concentration for modeled gas-phase ELVOCs, $NH_3$, and $H_2SO_4$. There is negligible model difference for ELVOC as concentration is steady state, and $NH_3$ is not consumed in the NPF function.



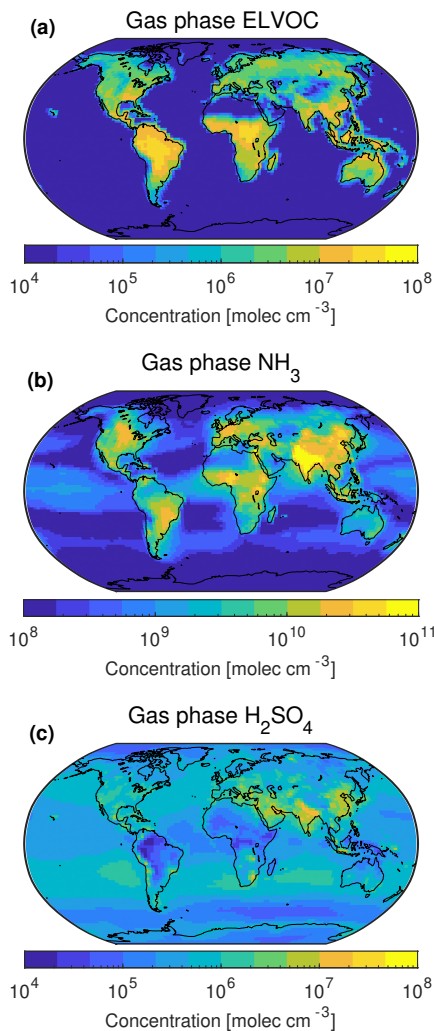

**Figure A3.** The mean near-surface gas phase concentrations of **(a)** ELVOC, **(b)** $NH_3$, and **(c)** $H_2SO_4$ for the EC-Earth3 control simulation.



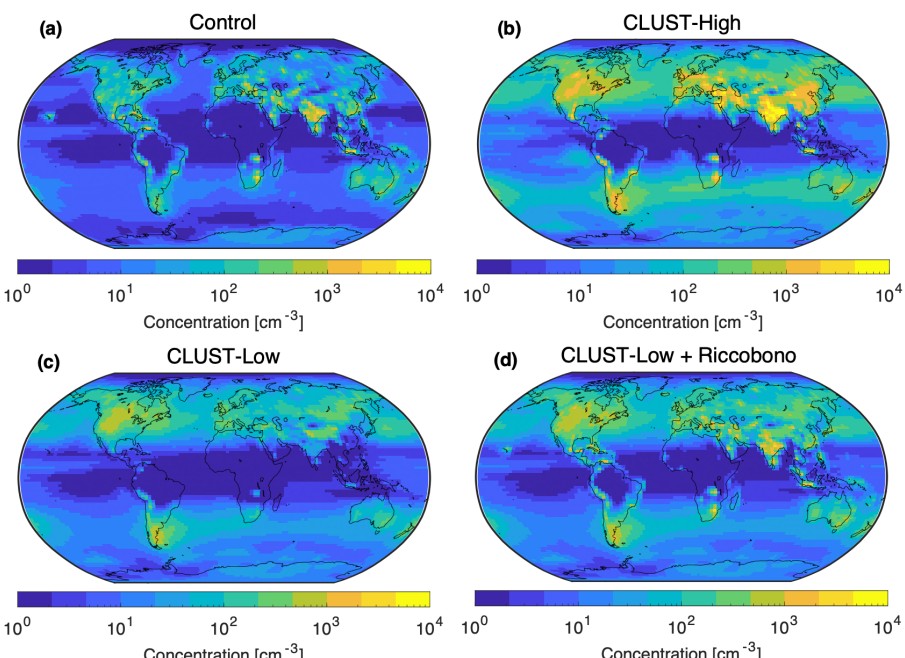

**Figure A4.** The mean near-surface aerosol nucleation mode (NUS) concentrations for the four simulations.





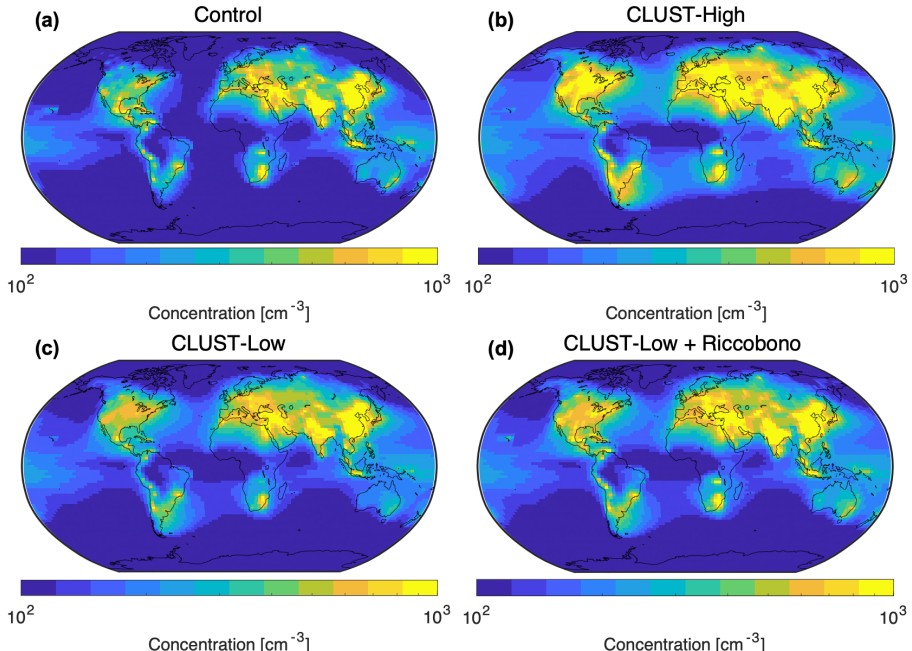

**Figure A5.** The mean near-surface aerosol Aitken mode (AIS) concentrations for the four simulations.





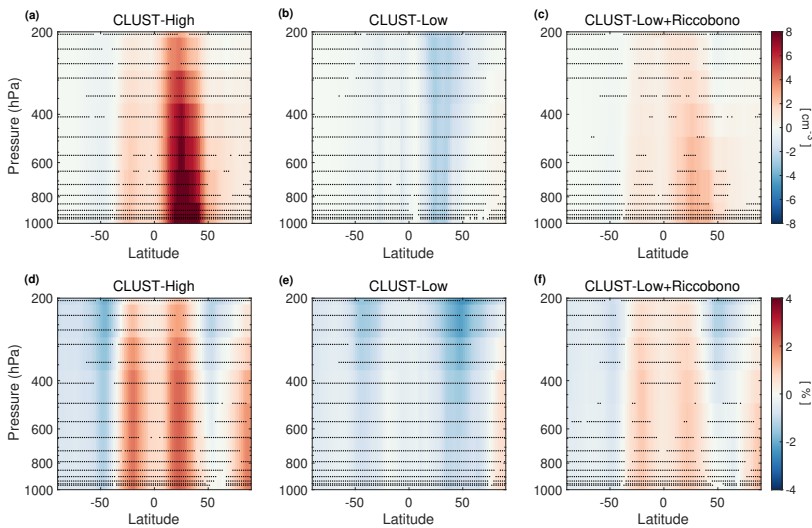

**Figure A6.** The zonal mean difference of the total aerosol number concentration of the soluble and insoluble accumulation and coarse mode (ACS, ACI, COS, and COI), with the absolute difference **(a)**, **(b)**, and **(c)**, and the relative difference **(d)**, **(e)**, and **(f)**, showing t-test significance as dots.



*Code and data availability.* Model code and descriptions for the adjusted EC-Earth3.3.4 TM5-MP version 1.2 with implemented CLUST look-up table is found at Svenhag (2024a). The model output datasets are found in Svenhag (2024b), with post-process scripts located in Svenhag (2024c). Codes for the J-GAIN v1.0 generator and the interpolator used for the CLUST look-up table in the experiments can be found at Yazgi and Olenius (2023a). Resources for the IPR lookup table can be found in Yu (2019). The DMPS and SMPS measurements
from the stations can be downloaded at https://ebas-data.nilu.no/Default.aspx.

*Author contributions.* CS, MS, TO, and PR designed the research idea, CS performed the simulations. TO and DY developed the CLUST model and CS performed the EC-Earth3 implementations. TO contributed to writing the CLUST model description. MS, PR, TO, and SB contributed to ideas in the discussion.

*Competing interests.* The authors declare that they have no conflict of interest.

*Acknowledgements.* This project was funded by the Swedish Research Council Formas (project no. 2018-01745-COBACCA) and the authors would like to thank the Swedish Research Council for Sustainable Development FORMAS (grant no. 2018-01745) for their financial support. This research has also been supported by Formas grant 2019-01433, the Swedish Research Council Vetenskapsrådet (grant no. 2019-05006 and 2019-04853), the Horizon Europe project AVENGERS (grant no. 101081322), the Crafoord foundation (grant no. 20210969). The computations and data handling were enabled by resources provided by the National Academic Infrastructure for Supercomputing in Swe-
den (NAISS) and the Swedish National Infrastructure for Computing (SNIC) at Tetralith partially funded by the Swedish Research Council through grant agreements no. 2022-06725 and no. 2018-05973. The authors thank the Technical Computing at Lund University (LUNARC) partially funded by the Swedish Research Council through grant agreements no. 2022-06725 and no. 2018-05973. Financial support from the European Union's Horizon 2020 research and innovation programme (project FORCeS under grant agreement No 821205), European Research Council (Consolidator grant INTERGRATE No 865799), and Knut and Alice Wallenberg Foundation (Wallenberg Academy Fel-
lowship project AtmoRemove No 2015.0162) are gratefully acknowledged. The authors thank Markku Kulmala and Pasi Aalto for DMPS measurements obtained at SMEARII, Hans Areskoug for DMPS measurements obtained at Aspvreten station, and Adam Kristensson for DMPS measurements obtained at Hyltemossa station. Additional thanks to David Picard, Jean-Marc Metzger, and Karine Sellegri for SMPS measurements from La Réunion station, and J.A. Casquero-Vera for SMPS measurements from Granada station. Markus Fiebig and Chris Lunder for DMPS measurements at Zeppelin station, Jean Putaud and Sebastiao Martins Dos Santos for measurements at Ispra station, and
Nicolas Bukowiecki and Urs Baltensperger for measurements at Zeppelin station. The authors appreciate the dedication, commitment, and effort of Randolph Borys, Gannet Hallar, Ian McCubbin, Dan Gilchrist, and Peter Atkins towards the long-term measurements of aerosols at Storm Peak Laboratory (SPL). We appreciate the data management support (including QA/QC) of Chris Rapp and Maria Garcia. Instrumentation at SPL used in this analysis was purchased via a grant AGS-0079486 and AGS-1040085 from the US National Science Foundation. The authors thank the National Institute of Meteorological Sciences (NIMS) funded by Developing Technology for Asian Dust and Haze
Monitoring and Prediction (KMA2018-00521) for SMPS measurements at Anmyeon-do. We acknowledge the use of particle size distribution data measured at the Izaña Observatory and kindly provided by the Meteorological State Agency of Spain within the frame of GAW-WMO





and ACTRIS programmes; the work of the technical staff and scientific in guarantee the quality of the data is recognized. The authors give great thanks to Roland Schrödner, Twan van Noije, Philippe le Seger, Klaus Wyser, and the EC-Earth Atmospheric Composition Working Group for their assistance. Additional appreciation and thanks to the ModEling the Regional and Global Earth System (MERGE) community and board.




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
