# Peer review of "Implementing detailed nucleation predictions in the Earth system model EC-Earth3.3.4: sulfuric acid-ammonia nucleation"

_EGUsphere, 2023_

## Author Response (AR1)

**Authors responses**

Carl Svenhag[1], Moa K. Sporre[1], Tinja Olenius[4], Daniel Yazgi[4], Sara M. Blichner[3],

Lars P. Nieradzik[2], and Pontus Roldin[1]

[1]Department of Physics, Lund University, Lund, Sweden

[2]Department of Physical Geography, Lund University, Lund, Sweden

[3]Department of Environmental Science and Analytical Chemistry, Stockholm University, Stockholm, Sweden

[4]Swedish Meteorological and Hydrological Institute, Norrköping, Sweden

*Corresponding author email: carl.svenhag@fysik.lu.se*

The authors are most grateful to the reviewers for the comments which have improved this manuscript. We present our responses to the questions and comments below. The comments from the reviewer are shown in bold text and our responses follow. We will answer the comments from Anonymous Referee 1 and 2 in this document.

**Anonymous Referee 1 (RC1):**

**Abstract:**

**A few lines are needed on the results of the model evaluation unless the last sentence is removed from the abstract.**

We have extended the description of model results and evaluation in the abstract. Line 15-18 now read as: "The modeled aerosol concentrations were compared to observed SMPS and DMPS measurements at various stations. In most cases, the new NPF predictions ($H_2SO_2$-$NH_3$) were performing better at stations where previous underestimations for aerosol concentrations occurred." This mentions the resulting different model performances in the compared regions with measured observations.

**Introduction:**

**Line 36: Reference CMIP (2022) is not listed.**

We have adjusted this, and this reference now exists. Reads as: Forster et al., 2021.

**Model description:**

**In order to make this a stand-alone paper, some description of the model setup in terms of forcings are required. What emissions are used for anthropogenic and natural emissions? For example, later in the results, marine NH3 is mentioned. These should be made available. In addition, the time period used in the study is represented by the SSPs in the CMIP6 context, so anthropogenic emissions are likely not standard CEDS emissions. Please clarify.**

We have clarified this by including an additional description and two references in Section 2.6 on the used emissions. As: "For input emissions, we use CMIP6 datasets (Feng et al., 2020), with "historical" for 2014 and the SSP3-7.0 scenario from 2015 onwards. This follows the emission standards in EC-Earth-AerChem simulations for CMIP6 given by Noije et al. (2021)."

**I would call section 2.4.1 as 2.5 as it is not linked to section 2.4.**

We have changed Section 2.4.1 to 2.5 per the reviewer's recommendation.

**Section 2.5: Please clarify if these are atmosphere only simulations, and briefly describe, for the general audience.**

We have clarified this by including a description of this in Section 2.1 Lines.92-94 which reads as: "the EC-Earth-AerChem configuration is atmosphere-only with sea-ice content and sea-surface temperature inputs from the AMIP reader. A more detailed description of all model couplings and components is given in van Noije et al. (2014, 2021)."

**Section 2.6: Please describe if any data filtering or gap filling procedure is applied to the observation data. What is the time resolution of the observations? Are there any cut-offs applied in evaluation? How are the measured bins mapped into modes?**

We have added further description of the time resolution of the observations in Section 2.7 (previously 2.6) Lines.185-1897 now reads as: "The particle number size distribution is measured using SMPS and DMPS instrumentation with 10-minute sampling intervals. The observations are then averaged to monthly mean values for uniformity with the model output."

We have also clarified that the modal model output from TM5 is mapped into sectional bins to be uniform with the measurements. Lines.206-207 Section 2.8 (2.7) now reads as: "The aerosol modal output from TM5-M7 is re-mapped into sectional bins to compare with the observations in units of dNdlogDp".

To address the gaps in the observation datasets we included a limitation for missing measured data with a description in Lines.187-189, this now reads as: "Months with measured data coverage below 50 % are excluded." This minimizes the effect of non-representative monthly averages.

The cut-offs for the observations are stated in Section 2.7 (2.6) Lines.188-189. This is further mentioned in the discussion as a limitation in the evaluation of the modeled nucleation mode.

**Section 2.7. Please clarify why the three-bottom layers have been used to represent the near surface.**

We have included a clarification in Section 2.8 (2.7) Lines.201-203 for why the 3 bottom layers are used. Stated as: "This is to capture the inhomogeneity seen in the bottom three model layers in Fig. 2a,d for sub-100 nm diameter  aerosols and H2SO4 (Fig. A2) caused by sink processes in the model surface layer."

**Results:**

**Section 3.1: At the near surface, the CLUST-Low formation rate is lower than the control case, globally. Is this also the case geographically, for example over emission hotspot areas and their downwind regions?**

We have now answered this question by adding an evaluation of this to Section3.1 Lines.212-214 as: "Fig. 1 shows that the near-surface formation rate for CLUST-Low is lower compared to the default control case, even in high-emission regions like China and India.". One exception is the mentioned marine NPF, where CLUST-low has formation rates, but the control case has none. It can also be noted that the nucleation scheme used in the control case (Riccobono) does not include any coagulation sink effect, unlike the new CLUST scheme. In highly polluted regions, particle concentrations are higher which decreases nucleation rates through scavenging of pre-nucleation clusters. This effect is included in the new scheme for the CLUST-Low case.

**Lines 235-236: Please explain why CLUST-Low case shows a decrease over the tropics in. the lower troposphere.**

This question was addressed in the original Manuscript in Section 3 Lines.215-217 as: "Some tropical regions have higher BVOC concentrations with lower NH3 and H2SO4 concentrations, so here ELVOC–H2SO4 near-surface nucleation is dominating in the model."

**Lines 238-239: Why do we see a spike in CLUST-High around 800-500 hPa?**

The spike at 500-800 hPa is explained by low temperatures, high ion production, and available $H_2SO_4$ and $NH_3$ from emissions in the northern mid-latitudes. This is discussed in section 3.1 Lines.239-243.

**Section 3.2:**

**I recommend the description of cut-off in observation to the methodology section, under section 2.6.**

This is addressed in the RC1 question for Section 2.6 above, where we refer to Section 2.6 Lines.188-189 where the cut-off for the instrumentations is stated.

**Section 3.3: Is it possible to also evaluate the model response in simulating liquid water content (LWP) and see if the response is as small as in liquid cloud radius?**

We have included a figure in the supplemental section (Fig. A7.) showing the total liquid cloud water column. A short result and discussion is added in Section 3.3. Lines.285-286 reads as: "The cloud liquid water content (seen in Fig. A7) increased in similar regions as the CDNC, and with 4 - 8 % increase globally."

**Line 267: Please clarify if the t-test is done for 95 percentile significance level.**

We have added this clarification for the mentioned t-test significance level.

**Discussion: I would also discuss briefly if the new scheme would also be applicable in the next version of EC-Earth.**

This we cannot say for sure as the next version EC-Earth4 is still under development. For later EC-Earth3 versions it can be made available as a module for download. This will be discussed in future meetings of the EC-Earth Aerosol working group. In principle, the scheme can be implemented in any model in which particle formation from gases is simulated.

**Anonymous Referee 2 (RC2):**

**l.5: Wording in "without loss of significant computational time". For me this sounds the opposite of what is meant. Could be "without significant additional computational burden/time/complexity"?**

We have changed this to a clearer wording for this sentence, and it now reads as: "In this study, we test and evaluate a new approach for improving the description of NPF processes in the ESM EC-Earth3 (ECE3) without significant additional computational burden.".

**l.14: Wording in "the changed nucleation only resulted in minor changes", perhaps change order of sentence.**

We have changed this sentence to make it clearer, and it now reads as: "Aerosol concentrations above 100 nm and the direct radiative effect (in Wm−2) showed only minor differences upon changing of the nucleation scheme."

**l.36: What reference is "CMIP6 2022" ?**

We have adjusted this, and this reference is now included. Reads as: Forster et al., 2021.

**l.37-48: Check consistency of BVOC vs. VOC. Perhaps you can only use BVOC here, or the only VOCs. But e.g. l. 42 "These BVOCs", when previous sentence can refer to also anthropogenic VOCs.**

We have modified this inconsistency for this section to clarify the difference between VOC and BVOC. Lines.42-45 now reads as: "In many Earth system models, BVOCs (and VOCs) are typically reduced to only two dominating species categorized by their volatility: semi-volatile (SVOC) and extremely low-volatile (ELVOC) (Sporre et al., 2020). These are two BVOC species that are primarily formed by the oxidation of two naturally emitted precursors isoprene and monoterpene."

**l.49: ESM defined here, but Earth System Models used already above. Perhaps define earlier.**

ESM is defined earlier in the abstract, and again in the Introduction when discussing model schemes typical to ESMs.

**l.79: Suggestion to rename "General" to more specific, e.g. "EC-Earth"**

We changed this section title to "EC-Earth3" following the reviewer's suggestion.

**l.93: Is there a reference for this "known issue"?**

This sentence on the "known issue" for this specific ECE3 version is mainly targeted towards readers from the EC-Earth-AerChem community, the development portal has a posted "issue" reference link on this, but this is accessible for members only.

**l.98: Perhaps indicate the standard deviations since they are different for different modes.**

We have added a description here for the standard deviations for all modes in Line.99-100 as: "The aerosol log-normal distribution has fixed geometric standard deviations of 1.59 for all modes except the coarse mode with 2.0."

**l.111: Bergman et al. (2021) seems to use yields directly from Jokinen et al. (2015), recommended to cite the original source of data.**

We have adjusted this reference per the reviewer's suggestion.

**l.165 and 167: Missing citation ("?").**

This reference is now corrected to Yu et al. (2019).

**l.187: Using monthly average data for aerosols. Is it possible that using monthly average values for mode diameters and mode concentrations causes biases in reconstructed aerosol size distributions (Fig.4)? For example, nucleation mode could have high concentrations after nucleation burst at ~1 nm, but as particles grow towards Aitken mode, their concentration is decreased (relocation to Aitken mode, coagulation). Reconstructing size distribution for 1-10 nm would be different if done with 1-hour (r,d) versus monthly average (<r>, <d>) data.**

Yes, we are aware that biases can arise when averaging is done on modal aerosol distributions as is done for Figure 4. We have previously investigated hourly timesteps and did not see any significant differences when plotting the modal structure averaged over longer periods compared to hourly data. We know the central diameter is rarely moving more than 5-10% from the median value for the nucleation mode. For the Aitken mode the median value moves more over time and the averaging will have a larger impact on this mode. We would expect the higher time resolution for the distribution would then increase the width of the median modes. Even so, the method of monthly averaging is not uncommon in model evaluation studies. As we do not have hourly TM5 output for these simulations we can only compare monthly outputs with the observational data.

**Fig.4: It is a good idea to include 25th percentile, but now some of the details are difficult to read. Even minor modifications might help, e.g. removing lines from shading edges and adjusting the opacity of shaded areas.**

We have adjusted Fig. 4 to have no shading edge lines and increased opacity to make the shaded percentiles clearer.